# An Improved Model of Single-Event Transients Based on Effective Space Charge for Metal–Oxide–Semiconductor Field-Effect Transistor

**DOI:** 10.3390/mi14112085

**Published:** 2023-11-11

**Authors:** Yutao Zhang, Hongliang Lu, Chen Liu, Yuming Zhang, Ruxue Yao, Xingming Liu

**Affiliations:** 1Key Laboratory for Wide Band Gap Semiconductor Material and Devices of Education Ministry, School of Microelectronics, Xidian University, Xi’an 710071, China; ytzhang_xd@163.com (Y.Z.); hllv@mail.xidian.edu.cn (H.L.); zhangym@xidian.edu.cn (Y.Z.); yaoruxue4305@126.com (R.Y.); 2SMiT Group Fuxin Technology Limited, Shenzhen 518000, China; liuxm@gwxeda.com

**Keywords:** CMOS inverter, current plateau effect, modeling method, single-event transient, soft error rate

## Abstract

In this paper, a single-event transient model based on the effective space charge for MOSFETs is proposed. The physical process of deposited and moving charges is analyzed in detail. The influence of deposited charges on the electric field in the depletion region is investigated. The electric field decreases in a short time period due to the neutralization of the space charge. After that, the electric field increases first and then decreases when the deposited charge is moved out. The movement of the deposited charge in the body mainly occurs through ambipolar diffusion because of its high-density electrons and holes. The derivation of the variation in electric field in the depletion region is modeled in the physical process according to the analysis. In combination with the ambipolar diffusion model of excessive charge in the body, a physics-based model is built to describe the current pulse in the drain terminal. The proposed model takes into account the influence of multiple factors, like linear-energy transfer (LET), drain bias, and the doping concentration of the well. The model results are validated with the simulation results from TCAD. Through calculation, the root-mean-square error (RMSE) between the simulation and model is less than 3.7 × 10^−4^, which means that the model matches well with the TCAD results. Moreover, a CMOS inverter is simulated using TCAD and SPICE to validate the applicability of the proposed model in a circuit-level simulation. The proposed model captures the variation in net voltage in the inverter. The simulation result obviously shows the current plateau effect, while the relative error of the pulse width is 23.5%, much better than that in the classic model. In comparison with the classic model, the proposed model provides an RMSE of 7.59 × 10^−5^ for the output current curve and an RMSE of 0.158 for the output voltage curve, which are significantly better than those of the classic model. In the meantime, the proposed model does not produce extra simulation time compared with the classic double exponential model. So, the model has potential for application to flow estimation of the soft error rate (SER) at the circuit level to improve the accuracy of the results.

## 1. Introduction

When an ionized particle strikes a semiconductor device, high-density electron–hole pairs are generated by material atoms due to the deposited energy along the track. These excessive carriers are collected by the sensitive area in the device, typically like negative-biased junctions in the device. A current pulse appears in the corresponding terminal, which may influence the normal working state of the circuit. This effect is called a single-event transient (SET). The space is full of high-energy ionized particles that are able to induce SET easily. With the scaling of manufacturing technology and the decrease in supply voltage, the SET becomes the major threat for spacecrafts and satellites [1,2,3]. Many investigations have been conducted to determine the physical mechanism and the influence factors of the SET. The modeling of single-event transients is one of the current research highlights, because the model is the bridge connecting the device level and circuit level and the model accuracy directly determines the result of circuit-level evaluation.

One of the methods of SET modeling is the use of an empirical model based on data fitting, which has been widely investigated in the past 40 years [4,5,6,7,8]. Messenger et al. proposed a double exponential model for the SET in 1982 [4]. This model has been widely used for a long time because of its concision and validity. As the demand for precision increases, other, similar models with higher accuracy have been presented, like the dual double exponential model [5,6,7]. Neural networks and machine learning have become popular nowadays, which has inspired researchers to apply this powerful tool in SET modeling. In 2021, an SET model based on machine learning regression was proposed [8]. However, for these models, numerical fitting is needed to extract the parameters in the model. To ensure the applicability and the accuracy of models, much data that cover actual working environment requirements are essential. The data under different conditions are difficult to obtain, which makes it difficult for this method to spread to new technology.

Another method is to model SETs by solving basic physics equations like Poisson’s equation and the continuity equation [9,10,11,12]. In 2018, Malherbe et al. presented a new model of SETs for full-depleted silicon-on-insulator (FDSOI) technology based on direct resolution of the drift-diffusion equations on a 1D grid [10]. In 2019, Aneesh et al. proposed a physics-based SET model by solving the 2D Poisson’s equation in DG (double-gate) MOSFETs [11]. Models like these have a relatively high accuracy and the ability to consider multiple influencing factors in actual applications. However, this method is difficult to use in general cases, because, in common cases, the equations may be highly complicated and the boundary conditions are difficult to determine, which makes them difficult to solve.

Another modeling method is based on analyzing the physical process of charges when an SET occurs [13,14,15,16,17,18,19,20]. Abadir et al. proposed a model to emulate SETs by taking into account the funnel effect in 2003 [13]. By calculating some critical physical parameters based on experience, the SET current was finally expressed. In 2010, an improved diffusion collection model was presented [14]. Artola et al. presented another improved model in 2011 [15]. The model accuracy was remarkably improved. However, these models do not capture the variation in drain voltage at the circuit level. The overshoot phenomenon would take place in the circuit simulation, which leads to relatively low accuracy.

In this paper, the physical process of deposited and moving charges is analyzed and described in detail when an SET occurs and a current model is proposed based on the analysis derived from the physics expression. The proposed model fits the SET results well with high accuracy at the device level. It captures the current plateau effect in the inverter and removes the overshoot problem using a voltage–control current source (VCCS) model. Moreover, the proposed model provides high accuracy in the circuit-level SET simulation with little extra time-consumption.

## 2. Modeling Method

Detailed TCAD simulation is conducted to investigate the movement of excessive carriers when an SET occurs. Several important electronic parameters are extracted to help analyze the process. And an SET model is derived after determining the mechanism of carriers’ movement.

As shown in Figure 1, a 2D NMOS is constructed using 45 nm CMOS technology, with the red dashed arrow indicating the ion track. The dimensions of simulated NMOS are gate length Lg = 45 nm, oxide thickness tox = 5 nm, source/drain doping of 1 × 10^20^ cm^−3^, p well (body in Figure 1) doping of 1 × 10^17^ cm^−3^. Figure 2 shows the simulated Id-Vg curves compared with process design kit (PDK) results from the open-source FreePDK 45 nm design kit. It can be seen that the simulation result fits well with the BSIM4 model.

The models used in the simulation are Shockley–Read–Hall (SRH), Auger recombination, bandgap-narrowing model, Fermi–Dirac statistics, Philips unified mobility model, high-field-saturation mobility model. The heavy ion model is introduced into the simulation to emulate SET. The heavy ion model becomes effective at 20 ps. The incident location is the center of the drain area, while the direction is perpendicular to the surface. The track length is 0.5 μm with gaussian profile along the radial direction. V_G_ and V_S_, the voltages of the gate and source terminal, are both set to 0 V, which is the sensitive mode for SET in NMOS.

Simulation is conducted with an LET of 0.5 pC/μm and a characteristic radius of 0.05 μm. The drain current curve is depicted in Figure 3. In order to understand the mechanism of carriers’ movement, a cutline is placed along the ion track shown in Figure 1, and several physical parameters are extracted at six different time points. The selected time points are 22 ps, 24 ps, 26 ps, 30 ps, 40 ps, and 100 ps, which comprehensively represent the carrier movement process, as indicated by the points and labels in Figure 3.

As shown in Figure 4, the space charge in the drain area is extracted. A red dashed line marks the metallurgical junction. From Figure 4, the space charge density decreases from 0 ps to 22 ps. Before ion injection, the space charge maintains its original state, which includes only ionized dopants. When the heavy ion is injected into the device, the space charge is partially neutralized as a result of the deposited charge, which leads to a decrease in space charge. Afterward, the deposited charge in the depletion region is moved out by the electric field. Due to the higher mobility of electrons, they are able to move out of the depletion region at a faster rate. During this process, the density of net positive charge increases and reaches its peak value when the electrons are completely removed, which occurs between 22 ps and 30 ps. And the holes are sequentially moved outward leading to a decrease in space charge from 30 ps to 100 ps.

The variation in the electric field is consistent with the space charge, as the electric field is directly related to the space charge in the depletion region. As shown in Figure 5, the electric field decreases from 0 ps to 22 ps due to the neutralization of the space charge. When the electrons are moved out of the depletion region, the electric field increases and reaches a peak as the space charge increases from 22 ps to 30 ps. Afterward, the electric field decreases and returns to its initial value from 30 ps to 100 ps.

Figure 6 illustrates the concentration of electrons and holes in the body along the ion track. Obviously, their concentrations are almost the same, and they are much larger than the doping concentration in the body region. The clear conclusion is that the excessive carriers in the body move through ambipolar diffusion, as mentioned in the previous study [15].

Based on the analysis above, the movement of the carriers can be divided into two stages: ambipolar diffusion and drift. In the initial stage, the carriers in the body move to the edge of the depletion region through ambipolar diffusion. The electrons are then swept into the drain area by the electric field in the depletion region, forming a current pulse in the drain terminal.

Figure 7 depicts a schematic illustration of the physical process following the injection of a heavy ion into devices. On the one hand, the deposited charge in the depletion region is moved out due to the electric field, which induces a variation in electric field in the process. On the other hand, the deposited charge in the body moves to the edge of the depletion region through ambipolar diffusion and is collected by the drain area.

The SET modeling process is divided into two parts based on the mechanism described above. The first part models the ambipolar diffusion of carriers from the body to the edge of the depletion region. The second part models the change in the electric field within the depletion region, which determines the drifting velocity of carriers. Referring to previous work [14], ambipolar diffusion can be calculated using Equation (1):(1)ne=∭LET(l)exp(−l2+x2+y24Dt−tτ)(4πDt)32dxdydl
where ne represents the carriers arriving at the border of the depletion region. LET is the linear energy transfer of the ion with the unit of pC/μm. *D* is the ambipolar diffusion coefficient, *τ* is the lifetime of carriers, and *q* is the elementary charge. dl is the elementary distance along the ion track. *dx* × *dy* = *dS*, where *dS* stands for the elementary area of the drain.

The drain–body junction can be treated as an abrupt n^+^-p junction. Therefore, the peak value of the electric field in the depletion region is calculated using (2):(2)$$E_{m}=\frac{q{N_{D}}^{\prime }W_{n}}{\varepsilon_{s}}$$
where *N_D_’* (*N_A_’*) represents the effective space charge, including ionized dopant and deposited charge in the depletion region, which reflects the variation in charge density. *ε_s_* represents the dielectric constant. *W_n_* is the depletion width in the n^+^ region, which is related to *N_D_’* and *N_A_’* and the space charge density on the body side. *W_n_* is expressed as (3).
(3)$$W_{n}=\frac{1}{{N_{D}}^{\prime }}\sqrt{\frac{2\varepsilon_{s}(\psi_{bi}-V_{0}){N_{A}}^{\prime }}{q}}$$
where *ψ_bi_* represents the built-in potential, and *V_0_* denotes the bias of the pn junction. In the calculation of the built-in potential in an abrupt junction, *ψ_bi_* is related to *N_D_’* and *N_A_’*.

According to the analysis, the key to characterizing the electric field is the space charge profile in the depletion region. From Section 2, it can be concluded that the space charge is determined by ionized dopants and induced electron–hole pairs in the depletion region. According to charge polarity, the density of space charge in the n^+^ region can be expressed as (4).
(4)$${N_{D}}^{\prime }=N_{D}+p(r,t)-n(r,t)$$
where *N_D_* is the density of ionized donors in the n^+^ region; *n(r, t)* and *p(r, t)* are the densities of excess electrons and holes, respectively; and *r* is the radial distance along the ion track. The excess carriers are moved out of the depletion region through drifting. To calculate the profile of the excessive carriers in the depletion region, the continuity equation (Equation (5)) needs to be solved.
(5)$$\frac{dp}{dt}=-\frac{1}{q}\nabla \cdot J_{p}-\frac{p}{\tau }$$

Referring to reference [4], the diffusion parallel to the ion track is negligible, which means that *−qDp*∇*p* equals 0. The generation rate is 0 because all electron–hole pairs are fully produced. Expression (5) is simplified according to these conditions. The density of initial excessive carriers is expressed as (6) in the heavy ion model.
(6)$$p_{0}=\frac{LET}{\sqrt{2\pi }qw_{t}}\exp\left[-{\left(\frac{r}{w_{t}}\right)}^{2}\right]$$
where *w_t_* is the characteristic radius of the ion track. By solving Equation (5) and substituting *p_0_* into it, the profile of excessive holes can be expressed as Equation (7).
(7)$$p(r,t)=\frac{LET}{q\sqrt{2\pi }w_{t}}\exp(-\frac{r^{2}}{w_{t}{}^{2}})\exp(-\frac{t}{\tau_{p}})$$
where *τ_p_* can be calculated by (8).
(8)$$\frac{1}{\tau_{p}}=\mu_{p}\frac{dE}{dx}$$
where *τ* is assumed to be much longer than the times of interest [4].

Using the same method, *n(r, t)* is calculated. And *N_D_’* is also fully expressed, as is *N_A_’*. Combining the above expressions, the electric field is finally calculated. Finally, the single-event transient current is calculated using Equation (9) by substituting the corresponding value.
(9)$$I=q\mu_{n}E_{m}n_{e}$$

In the calculation, we assume that the electric field is constant in the depletion region along the ion track. However, the electric field varies at every point. Therefore, the final expression is multiplied by an appropriate factor B to adjust the amplitude of the transient current.

The model equation finally becomes (10) and (11).
(10)$$I=B\frac{\mu_{n}LET^{2}}{2\sqrt{2\pi }N_{D}w_{t}}\sqrt{\frac{(\psi_{bi}-V_{0})N_{A}}{q\varepsilon_{s}}}erf(\frac{x_{m}}{\sqrt{{4Dt}^{\prime }}})\left[\exp(-\frac{t^{\prime }}{\tau_{p}})-\exp(-\frac{t^{\prime }}{\tau_{n}})\right]\left[erf(\frac{l_{m}}{\sqrt{{4Dt}^{\prime }}})-erf(\frac{l_{0}}{\sqrt{{4Dt}^{\prime }}})\right]+B\cdot I_{0}$$
(11)$$I_{0}=\mu_{n}LET\sqrt{\frac{q(\psi_{bi}-V_{0})N_{A}}{4\varepsilon_{s}}}erf(\frac{x_{m}}{\sqrt{{4Dt}^{\prime }}})\left[erf(\frac{l_{m}}{\sqrt{{4Dt}^{\prime }}})-erf(\frac{l_{0}}{\sqrt{{4Dt}^{\prime }}})\right]$$
where *t’ = t − t*_0_, and *t*_0_ represents the delay time for the generation of electron–hole pairs. *x_m_* represents the radius of electron–hole pairs in the body, *l_m_* is the length of the ion track, and *l*_0_ is the junction depth of the drain.

## 3. Model Validation and Discussion

An SET model is proposed in the last section. The modeling process is summarized. The variation in the electric field is caused by the movement of excessive carriers in the depletion region. So, the first step is to calculate the profile of the excessive charge in the depletion region. The second step is to calculate the depletion width and actual space charge density. The electric field can be expressed in terms of the depletion width and the space charge density. The ambipolar diffusion within the body is calculated using Equation (1). Finally, the SET current is characterized using Equation (9) by substituting the corresponding items calculated earlier.

The SET current results from Equation (10) for different LETs are plotted in Figure 8 and compared with TCAD simulation results. With the increase in LET, the deposited charge also increases, leading to a higher current amplitude and longer collection time. The density of excess carriers in the depletion region increases, prolonging the process of electric field recovery. The duration of the current increase is clearly longer, as it is not the FWHM of the SET current, but rather the time it takes to recover to a certain current level. From the proposed model, the amplitude of the SET current is proportional to the square of the LET, which is consistent with the simulation results. The RMSE between the proposed model and TCAD simulation is 3.7 × 10^−4^, even when the LET is 1 pC/μm. The RMSE decreases at lower LET values. The comparison between the model results and TCAD results indicates that the model aligns well with the simulation data.

Figure 9 illustrates the impact of drain voltage on the SET current. The simulation results are also depicted in Figure 9. As the drain voltage increases, the peak current value also increases due to the relationship between the electric field in the depletion region and the drain reverse bias. The proposed model predicts this trend due to the inclusion of *V*_0_, which represents the drain voltage in the expression. The electric field in the depletion region increases with the increase in reverse bias *V*_0_. The drift velocity of electrons in the depletion region also increases. In Figure 9, the model demonstrates comparable results to the TCAD simulation, and the RMSE results further confirm the strong agreement between the model and the TCAD simulation.

In Figure 10, the proposed model is compared with the classic double exponential model in the SET current under an LET of 0.6 pC/μm and the V_D_ of 0.2 V. The RMSE between the double exponential model and TCAD simulation is 2.09 × 10^−3^, which is 6.24 times larger than that of the proposed model. It is evident that the proposed model has a significantly higher accuracy compared to the double exponential model. The double exponential model only fits the simulation data in the early stage, while the proposed model matches well through almost the entire time range. The proposed model is based on a new explanation of the SET mechanism, which is effective through the entire process. The carriers’ movement is divided into two stages, which is reasonable for the entire process. This is a beneficial enhancement for SET modeling.

When the incident occurs in the channel region, the model derivation is similar to the one shown in the previous section, with corrections made to the electric field. In this case, the ion track does not pass through the sensitive area, which is the drain–body junction. An inference can be made that the electric field in the drain area remains constant during the occurrence of an SET, which is consistent with its initial electric field. Therefore, the expression of the electric field in this case differs from that when the incident location is in the drain area. After that, the movement of the deposited charge is still considered to be ambipolar diffusion. The difference is that the distance between the deposited charge and the collecting area is greater. So, based on the analysis, it is concluded that the expression for the electric field needs to be corrected when the incident location is in the channel region.

Furthermore, we validate the application of the proposed model in a circuit-level simulation. An inverter is simulated using TCAD and SPICE. The Vout-Vin curves are depicted in Figure 11. We can observe that the results are nearly identical in TCAD and SPICE. Based on these two inverters, the circuit-level SET simulation is conducted with an input state of “0”, indicating that the NMOS in the inverter is sensitive to SETs. The SET current is applied to the drain terminal of the NMOS. The method of introducing SET current into circuit simulation is similar to the classic approach used in Verilog-A. The difference is that the proposed SET model is emulated by a Voltage-Controlled Current Source (VCCS) where the control voltage is Vds. The proposed model provides the capability to sense the variation in drain voltage, enabling it to be modeled as a VCCS. Meanwhile, an inverter with an independent source based on the double exponential model is also simulated in SPICE for comparison.

Figure 12 shows the output voltage simulated in SPICE and TCAD under an LET of 1 pC/μm. The simulation results, based on the double exponential model, are also depicted in Figure 12. The simulated output voltage curve, based on the independent source, indicates that the output level is pulled to the negative side, as shown by the solid black line in Figure 12. The simulation results using the proposed model do not exhibit a similar phenomenon, indicating that the overshoot problem is effectively resolved compared to the double exponential model. Compared with the TCAD simulation results, the proposed model yields a more accurate output voltage curve than the double exponential model. Figure 13 displays the output current curves from these two models, compared with the TCAD simulation. The current curve using the proposed model exhibits an obvious plateau effect, consistent with the TCAD simulation and actual physical phenomena. The current based on the independent source exhibits a much higher amplitude than the proposed model. This indicates that the net voltage cannot significantly influence the waveform of the SET current in circuits, making it markedly different from the waveform in the device-level simulation.

Upon comparing the current and voltage curves, the simulation results obtained using the proposed model align more closely with the TCAD simulation results than those obtained using the double exponential model. The results from the double exponential model indicate a clear “overshoot” in the output voltage curve. The main reason is that the double exponential model does not account for the impact of net voltage on the SET process, which fluctuates within this time range. The double exponential model is constructed as an independent source due to this characteristic inherent in the double exponential model itself. As a result, the “overshoot” phenomenon occurs. The proposed model addresses this issue by considering the net voltage. In the proposed model, the net voltage influences the space charge region through the entire process, as clearly demonstrated in the results. Based on this enhancement, the proposed model is designed as a dependent model capable of detecting variations in net voltage. In the results, the output voltage does not exhibit an “overshoot” phenomenon, and the current shows a plateau, which is validated through TCAD simulation.

Table 1 presents a comparison between the proposed model and the double exponential model. The simulation time for the double exponential model and the proposed model is nearly identical, indicating that the proposed model does not introduce additional simulation complexity. Compared with 3.96 × 10^−3^ using the double exponential model, the RMSE of the output current using the proposed model is 7.59 × 10^−5^, which represents a significant improvement. The RMSE of the output voltage using the proposed model is 3.35 times lower than that using the double exponential model. Another important parameter to consider is the pulse width. The relative error in pulse width using the proposed model is 23.5% compared to the TCAD result, which is more accurate than that using the double exponential model.

## 4. Conclusions

This paper analyzes and describes the physical process of deposited charge in detail and presents an improved SET current model for MOSFETs. By analyzing the profile of the deposited charge in the depletion region, the variation in the electric field can be determined. The difference between electrons and holes leads to the electric field initially increasing and then decreasing after a brief period, caused by the partial neutralization of space charge. The movement of excessive charge in the body is primarily ambipolar diffusion, which is consistent with previous research. Based on the analysis, a physical model is developed to simulate the SET current. Validation is conducted, and the results indicate that the proposed model is highly accurate. The root-mean-square error (RMSE) between the simulation and the model is less than 3.7 × 10^−4^, indicating a strong match with the TCAD results. In particular, the circuit-level simulation demonstrates that the proposed model accurately characterizes the current plateau effect. The overshoot problem caused by the independent source is resolved by the proposed model due to its ability to detect the variation in drain voltage in the actual circuit. A comparison is conducted between the proposed model and the classic double exponential model. The results show that the proposed model yields an RMSE of 7.59 × 10^−5^ for the output current curve and an RMSE of 0.158 for the output voltage curve, which are significantly better than those of the classic model. The relative error in pulse width between the TCAD and the proposed model is 23.5%. Meanwhile, the proposed model does not require additional simulation time compared to the classic double exponential model. So, the model has potential for application to flow estimation of the soft error rate (SER) at the circuit level to improve the accuracy of the results.

## Figures and Tables

**Figure 1 micromachines-14-02085-f001:**
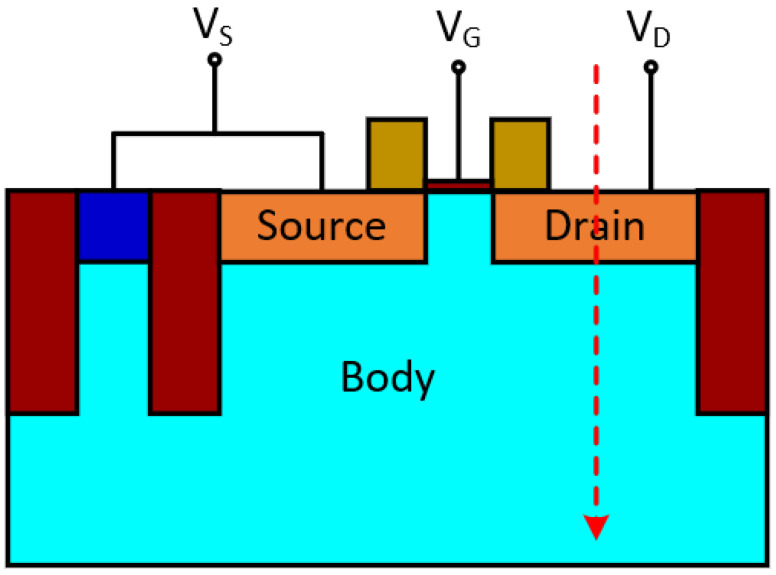
NMOS structure schematic using 45 nm CMOS technology. The electrode of body is on the top of bulk, which is connected with source.

**Figure 2 micromachines-14-02085-f002:**
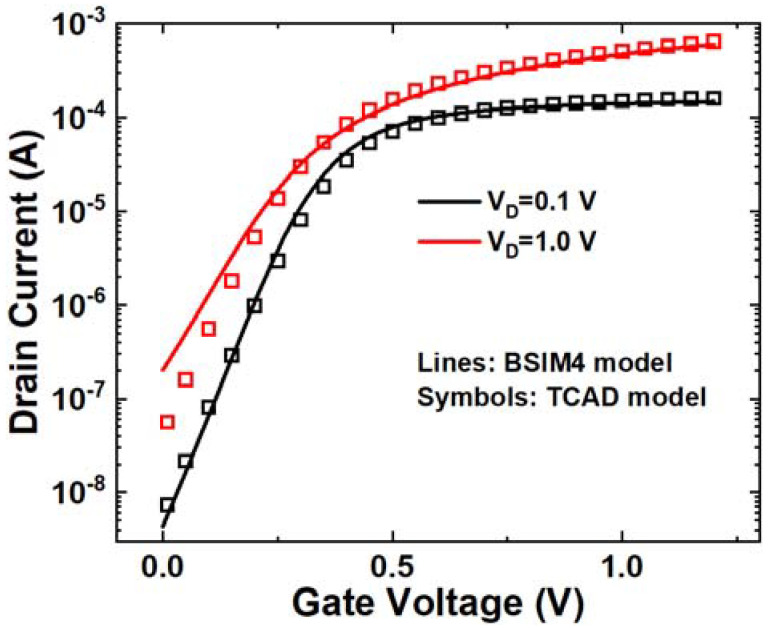
Simulated Id-Vg curves compared with PDK results.

**Figure 3 micromachines-14-02085-f003:**
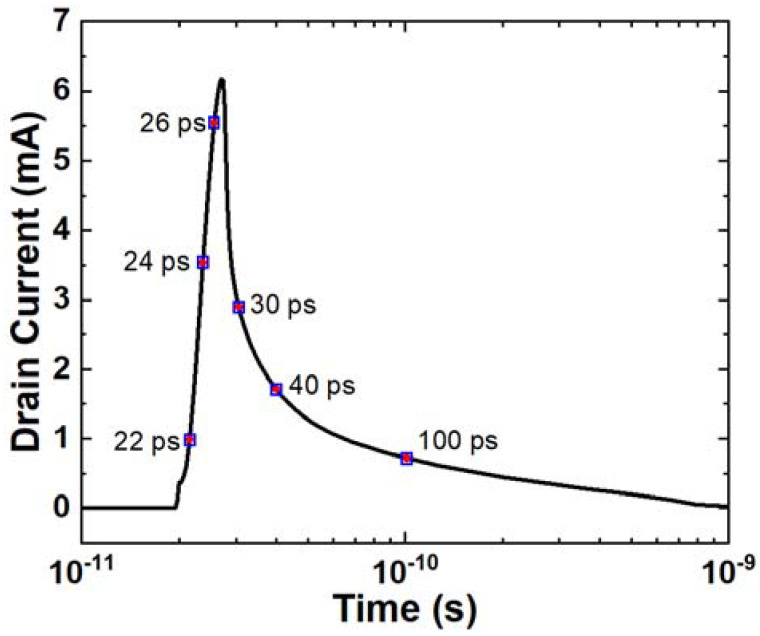
Single-event transient current in drain terminal with an LET of 0.5 pC/μm and a characteristic radius of 0.05 μm. Six points of time that cover almost the whole process of SET are chosen to extract the electronic parameters, which are marked using points and texts.

**Figure 4 micromachines-14-02085-f004:**
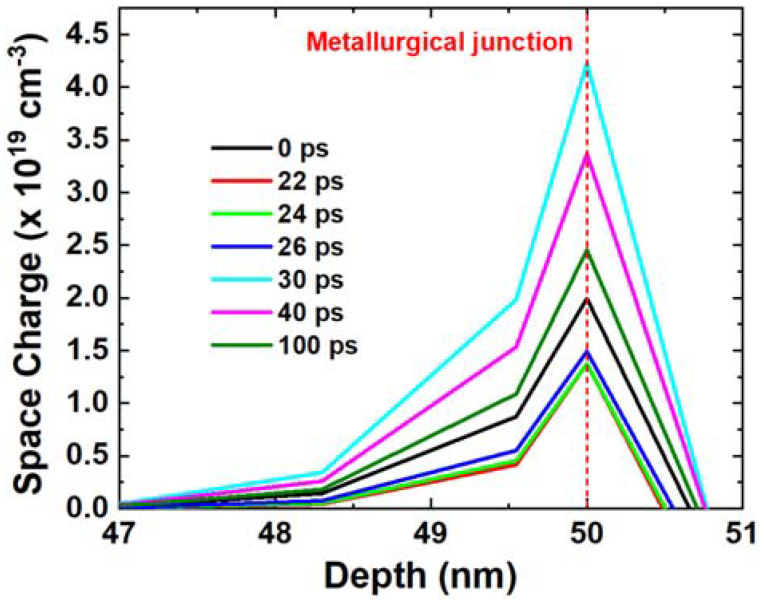
Space charge around metallurgical junction extracted along the cutline at different time points. The red dashed line indicates the metallurgical junction.

**Figure 5 micromachines-14-02085-f005:**
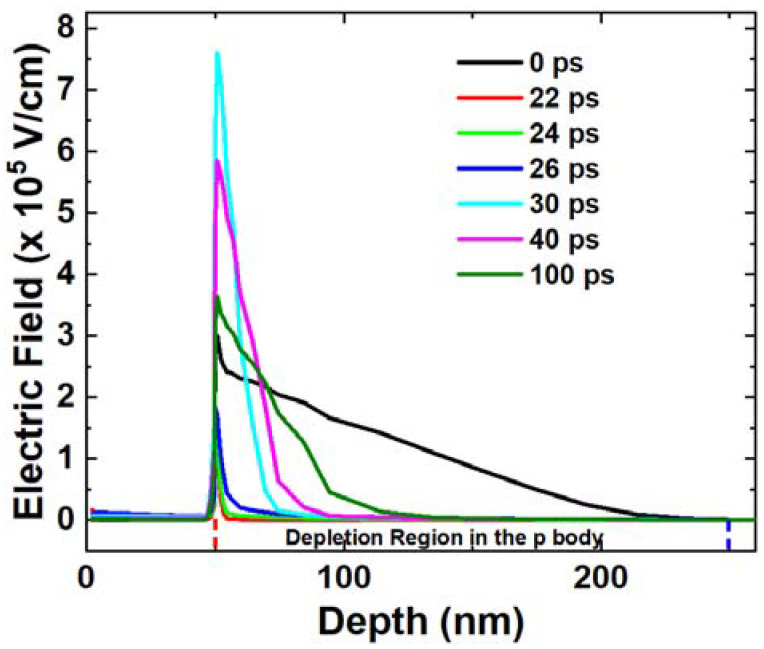
Electric field extracted along the cutline at different time points. The red dashed line indicates the metallurgical junction and the blue dashed line indicates the border of original depletion region.

**Figure 6 micromachines-14-02085-f006:**
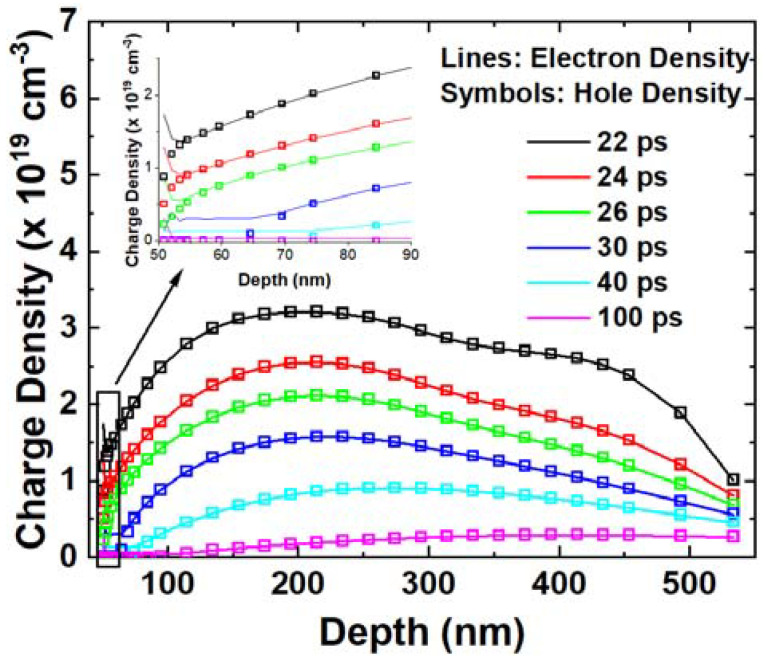
Electron and hole densities in the body are extracted along the cutline at different time points. The inset picture indicates the carriers’ density close to the metallurgical junction.

**Figure 7 micromachines-14-02085-f007:**
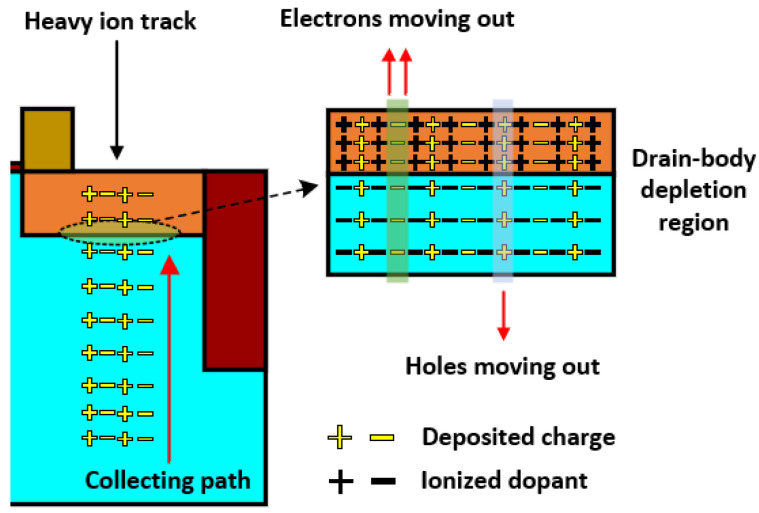
The schematic illustration of the physical process after heavy ion injects into devices.

**Figure 8 micromachines-14-02085-f008:**
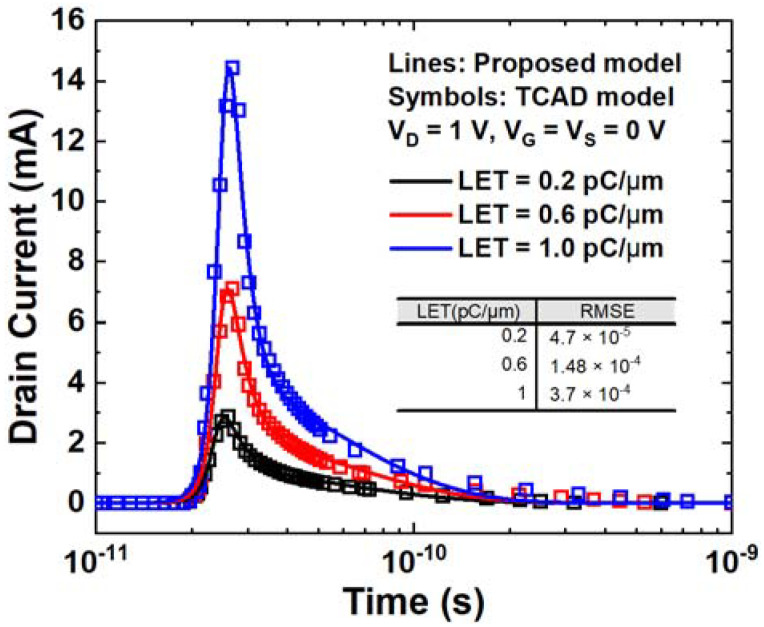
SET current curves versus time for different LETs. The voltages setting is V_G_ = V_S_ = 0 V, V_D_ = 1 V.

**Figure 9 micromachines-14-02085-f009:**
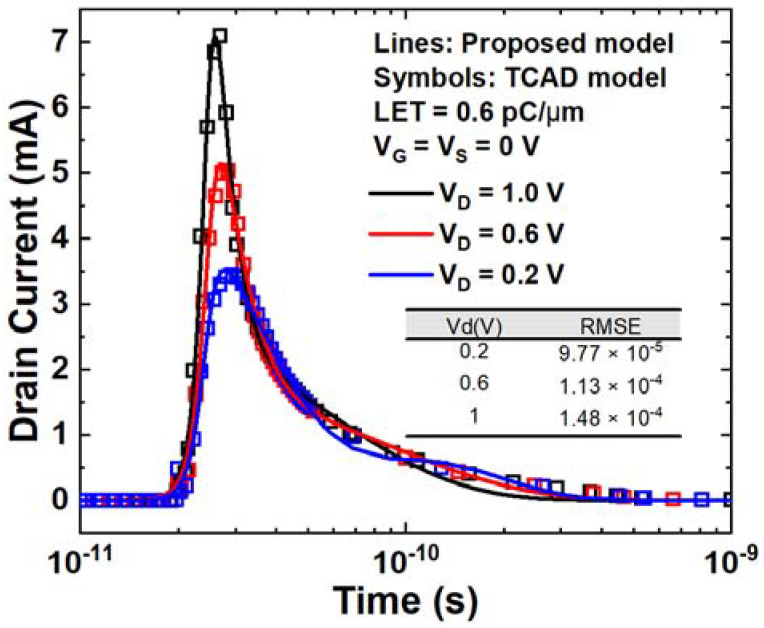
SET current curves versus time for different drain voltages. The voltages setting is V_G_ = V_S_ = 0 V and the LET is 0.6 pC/μm.

**Figure 10 micromachines-14-02085-f010:**
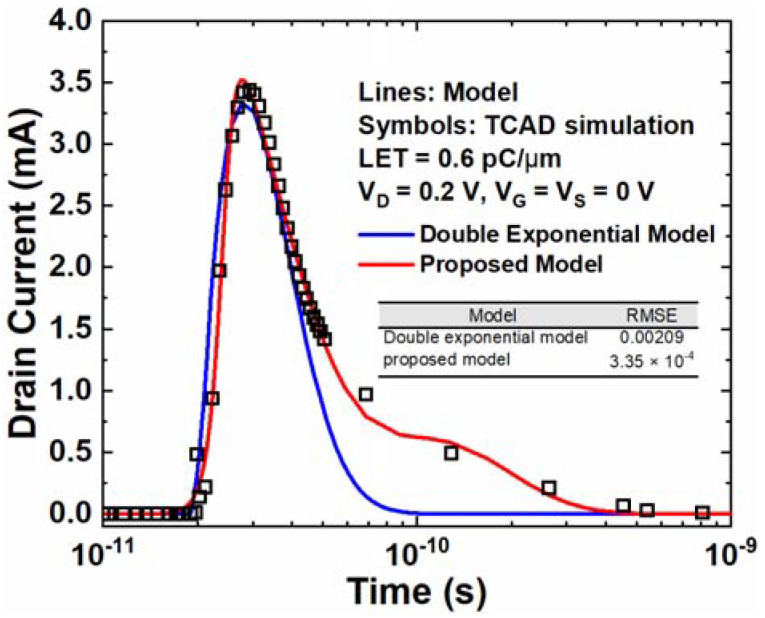
SET current comparison between TCAD, proposed model, and double exponential model. The voltage setting is V_G_ = V_S_ = 0 V and V_D_ = 0.2 V and the LET is 0.6 pC/μm.

**Figure 11 micromachines-14-02085-f011:**
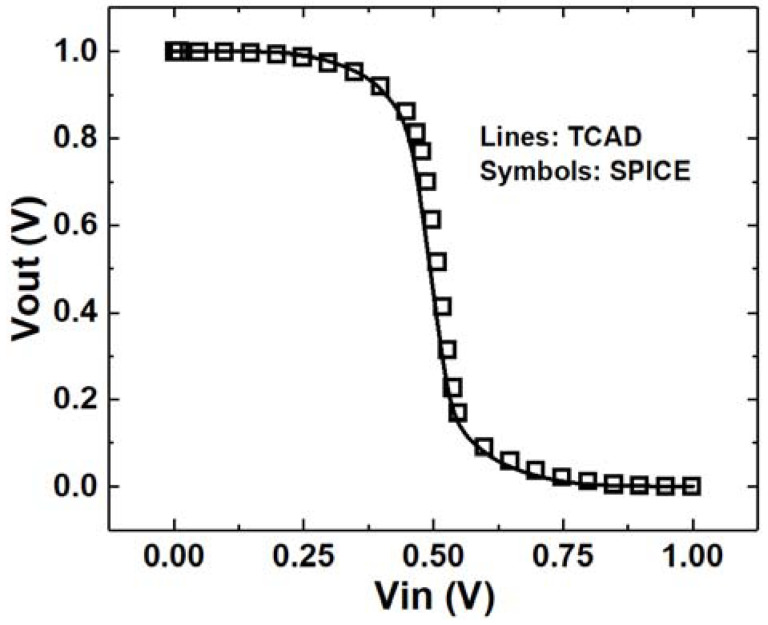
Vout-Vin results in TCAD and SPICE.

**Figure 12 micromachines-14-02085-f012:**
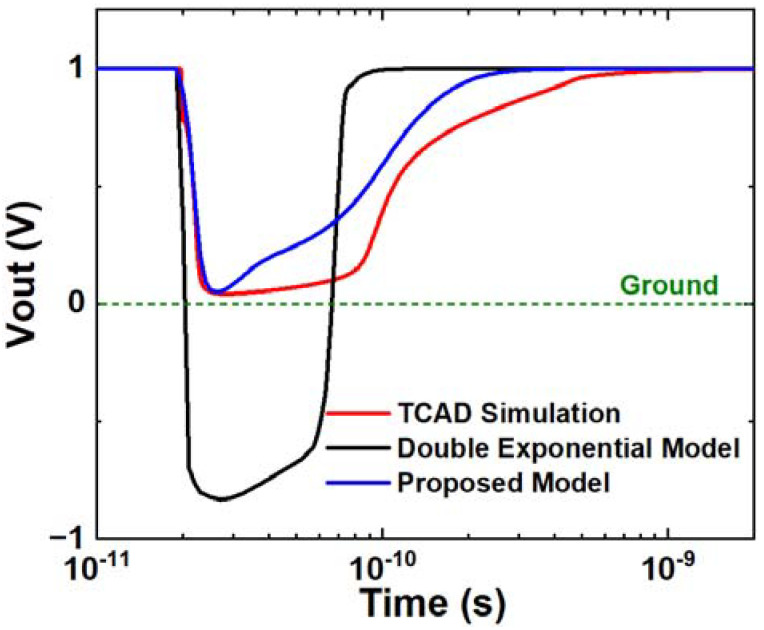
Output voltage comparison among proposed model, double exponential model, and TCAD simulation in inverter SET simulation. The input state is Vin = 0 V and the LET is 1 pC/μm. A green dashed line is drawn in the figure indicating the 0 V voltage level.

**Figure 13 micromachines-14-02085-f013:**
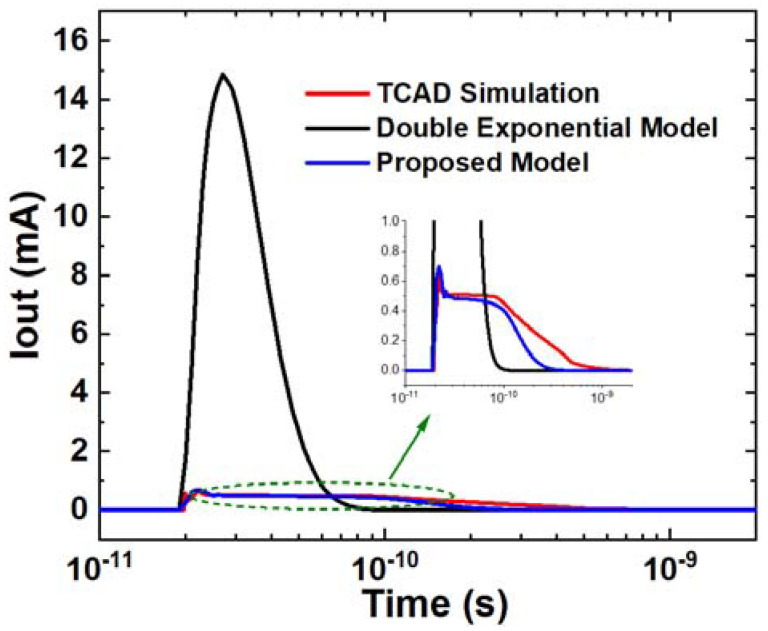
Output current comparison among proposed model results, double exponential model, and TCAD in inverter SET simulation. The input state is Vin = 0 V and the LET is 1 pC/μm. The green dashed box indicates the current plateau effect. The inset figure is an enlarged view of simulation results based on proposed model and TCAD.

**Table 1 micromachines-14-02085-t001:** Comparison between the proposed model and the classic double exponential model. The baseline is the simulation results of the inverter in TCAD under Vin = 0 V, LET = 1 pC/μm, and VDD = 1 V.

	Double Exponential Model	Proposed Model
Simulation time	0.065 ms	0.064 ms
RMSE ^1^ of output current	3.96 × 10^−3^	7.59 × 10^−5^
RMSE of output voltage	0.529	0.158
Relative error of pulse width	52.2%	23.5%

^1^ Root-mean-square error.

## Data Availability

Data are contained within the article.

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
