# Peer review of "An Improved Model of Single-Event Transients Based on Effective Space Charge for Metal–Oxide–Semiconductor Field-Effect Transistor"

_micromachines, 2023, doi:10.3390/mi14112085_

Round 1

Reviewer 1 Report

Comments and Suggestions for Authors

This paper proposed a more accurate model for SET effect in MOSFET without producing extra cost of time. Based on the analysis, the movement of excessive carriers is divided into two stages, ambipolar diffusion and drift, which is suitable for the whole time-range. The conclusion is reasonable. There are some questions need to be addressed.

1.     As for Figure 4, could you please explain the phenomenon more clearly from the point view of Poisson equation or microcosmic other than resistance. If necessary, you can show it in schematic drawing of charge distribution and moving.

2.     Is this model suitable for the situation that the incident location is channel region? For this case, does the model need to be corrected by the lateral electric field. Please discuss it.

3.     The introduction for existing methods of SET modeling is too prolix. Please summarize appropriately and simplify it.

4.     The format of reference is inconformity, like “IEEE TRANSACTIONS ON NUCLEAR SCIENCE”, “IEEE Trans. Nucl. Sci.”.

Comments on the Quality of English Language

The expression of absctract need  to be more refined.

Author Response

Response to Reviewer 1 Comments

Dear reviewer,

We would like to thank you for your efforts in reviewing our manuscript titled "An Improved Model of Single-Event Transients Based on Effective Space Charge for MOSFET"(2690903), and providing many helpful comments and suggestions, which are valuable for the revision and improvement of our study.

Based on that we significantly modified the document. The detailed point by point response is added as a separate word document. The abstract is also refined.

Thank you again for your comments and suggestions.

Response to Reviewer Comments:

Point 1: As for Figure 4, could you please explain the phenomenon more clearly from the point view of Poisson equation or microcosmic other than resistance. If necessary, you can show it in schematic drawing of charge distribution and moving.

Response 1:  Thank you for your constructive comments. We have made modifications to Modeling method section by adjusting the sequence of original Figure 4 (Figure 5 in revised version) and Figure 5 (Figure 4 in revised version). The change of space charge is analyzed at first. The electric field in depletion region is related to the concentration of space charge. So, from the analysis of space charge, the variation of electric field can be explained. The modifications of figures and texts are highlighted in revised manuscript.

Point 2: Is this model suitable for the situation that the incident location is channel region? For this case, does the model need to be corrected by the lateral electric field. Please discuss it.

Response 2: Thank you for your constructive comments. We have made modifications to Model Validation and Discussion section by adding the analysis of the case of incident location in channel region, which are highlighted in the revised manuscript.

Point 3: The introduction for existing methods of SET modeling is too prolix. Please summarize appropriately and simplify it.

Response 3: Thank you for your advice. We have made modifications to this issue by simplifying the introduction for existing methods of SET modeling which are highlighted in the Introduction section.

Point 4: The format of reference is inconformity, like “IEEE TRANSACTIONS ON NUCLEAR SCIENCE”, “IEEE Trans. Nucl. Sci.”.

Response 4: Thank you for your advice. We have made modifications to this issue which are highlighted in the Reference section.

Reviewer 2 Report

Comments and Suggestions for Authors

This manuscript introduced a single-event transient model based on effective space charge for MOSFET considering of multiple factors such as LET, drain bias and doping concentration. The profile of several electrical parameters in a sample NMOS was analyzed when SET happens. Overall, the manuscript was well organized and provided highly supportive experimental results. However, there are major comments that authors should revise through the overall manuscript.

1.      Authors proposed the newly invented simulation methodology. However, the detailed information is not provided. Please suggest the comparison table of conventional modelling tools comparing with this research results based on the accuracy and the simulation time.

2.      Although, the data is presented as various graphs, the detailed information of results are not enough. The numerical value should be produced for reasonable understanding.

3.      Please provide the schematic illustration of the phenomenon of major reaction occurred on the transistor followed the explanation in this paper.

As a result, I can recommend this manuscript for publication after Major Revision in Micromachines.

Comments on the Quality of English Language

The overall manuscript should be revised for the submission to Micromachines. The minor editing of paper is essential.

Author Response

Response to Reviewer 2 Comments

Dear reviewer,

We would like to thank you for your efforts in reviewing our manuscript titled "An Improved Model of Single-Event Transients Based on Effective Space Charge for MOSFET"(2690903), and providing many helpful comments and suggestions, which are valuable for the revision and improvement of our study.

Based on that we significantly modified the document. The detailed point by point response is added as a separate word document.

Thank you again for your comments and suggestions.

Response to Reviewer Comments:

Point 1: Authors proposed the newly invented simulation methodology. However, the detailed information is not provided. Please suggest the comparison table of conventional modelling tools comparing with this research results based on the accuracy and the simulation time.

Response 1:  Thank you for your constructive comments. We have replaced original Figure 13 with Table 1 to compare the classic model and proposed model based on the accuracy and simulation time, which is highlighted in Model Validation and Discussion section.

Point 2: Although, the data is presented as various graphs, the detailed information of results are not enough. The numerical value should be produced for reasonable understanding.

Response 2: Thank you for your constructive comments. We have made modifications to Figure 8, 9 and 10 in revised manuscript by calculating RMSE between model results and simulation results. The illustration texts are also modified. The modifications are highlighted in Model Validation and Discussion section.

Point 3: Please provide the schematic illustration of the phenomenon of major reaction occurred on the transistor followed the explanation in this paper.

Response 3: Thank you for your constructive comments. We have made modifications to this issue by adding a schematic illustration Figure 7 and the whole process is completely introduced based on Figure 7, which are highlighted in revised manuscript.

Round 2

Reviewer 1 Report

Comments and Suggestions for Authors

All my questions have been addressed in the revised manuscript. No other comments.

Reviewer 2 Report

Comments and Suggestions for Authors

The authors edited as the reviewer mentioned and the manuscript was updated with revised data in the revised version.

 As a result, I recommend this manuscript for publication in Micromachines.

Comments on the Quality of English Language

Minor editing through overall the manuscript is essential before publication in Micromachines.